

# A new specimen of *Sinopterus dongi* (Pterosauria, Tapejaridae) from the Jiufotang Formation (Early Cretaceous, China)

Caizhi Shen[1,*], Rodrigo V. Pêgas[2,*], Chunling Gao[1], Martin Kundrát[3], Lijun Zhang[4], Xuefang Wei[5] and Xuanyu Zhou[6,7,8]

[1] Dalian Natural History Museum, Dalian, Liaoning, China

[2] Laboratório de Paleontologia de Vertebrados e Comportamento Animal, Federal University of ABC, São Paulo, Brazil

[3] Evolutionary Biodiversity Research Group, PaleoBioImaging Lab, Center for Interdisciplinary Biosciences, Technology and Innovation Park, Pavol Jozef Safárik University, Kosice, Slovak Republic

[4] Hainan Tropical Ocean University, Sanya, Hainan, China

[5] Institute of Geology, Chinese Academy of Geological Sciences, Beijing, China

[6] School of Sciences, Hokkaido University, Sapporo, Japan

[7] Hokkaido University Museum, Hokkaido University, Sapporo, Japan

[8] Beipiao Pterosaur Museum of China, Beipiao, Liaoning, China

[*] These authors contributed equally to this work.

Corresponding author
Xuanyu Zhou,
xyzhou@elms.hokudai.ac.jp

## ABSTRACT

The Tapejarinae are edentulous pterosaurs that are relatively common in Cretaceous continental deposits in South America, North Africa, Europe, and China (mostly Early Cretaceous). The Chinese Jiufotang Formation is particularly rich in tapejarine specimens, having yielded over 10 described specimens and dozens of undescribed ones. For the Jiufotang Formation, a total of seven nominal tapejarid species and two genera have been proposed. Some debate exists over how many of those are valid or, alternatively, sexual or ontogenetic morphs of fewer (or even a single) species. Despite the abundance of specimens and the relevant taxonomic problems involved, detailed revisions of the matter are still lacking. This is partly due to the relatively scarce knowledge on the comparative osteology of the *Sinopterus* complex, which is hampered by the fact that most specimens have been only preliminarily described. In this contribution, we present a new postcranial specimen, D3072, which we attribute to the type-species of the genus, *Sinopterus dongi*. This new specimen helps shed some new light in the osteology of *Sinopterus dongi*, hopefully serving as a basis for future comparative studies involving further specimens and other proposed species and, subsequently, taxonomic revisions.

## INTRODUCTION

The Tapejarinae (*sensu Kellner & Campos, 2007*) are a peculiar clade of Cretaceous edentulous pterosaurs of the group Azhdarchoidea (Pterodactyloidea, Eupterodactyloidea),

defined as the most inclusive clade comprehending *Tapejara wellnhoferi* but not *Thalassodromeus sethi* (*Kellner & Campos, 2007*). They are characterized by their short, downturned rostra and sagittal premaxillary and dentary crests that start on the rostral region of the skull (*Kellner & Campos, 2007*). They comprise over 10 species (up to 19 proposed species), spanning from the Barremian to the Santonian; with records from Brazil, Morocco, Europe and China (*Kellner & Campos, 2007*; *Vullo et al., 2012*; *Andres, Clark & Xu, 2014*; *Pêgas, Leal & Kellner, 2016*; *Martill et al., 2020*).

In China, tapejarines are a common element of the famous Jehol Biota (*e.g., Wang & Zhou, 2006*). From the Yixian Formation (late Barremian—early Aptian), a single species, represented by two specimens, has been described: *Eopteranodon lii* (*Lü & Zhang, 2005*; *Andres & Ji, 2008*; *Vullo et al., 2012*). It is, however, in the Jiufotang Formation (early Aptian) that a great abundance of tapejarines is found. A total of 12 Jehol tapejarine specimens have been formally reported in the literature (*Wang & Zhou, 2003a*; *Li, Lü & Zhang, 2003*; *Lü & Zhang, 2005*; *Lü & Yuan, 2005*; *Lü et al., 2006a*; *Lü et al., 2006b*; *Lü et al., 2006c*; *Lü et al., 2007*; *Lü et al., 2016*; *Liu et al., 2015*; *Zhang et al., 2019*). Under the accounts of *Wu, Zhou & Andres (2017)*, at least further nine undescribed specimens are currently deposited in Paleontological Museum of Liaoning, and further four in Shandong University of Science and Technology. According to our observations, further tens of specimens can be found in the collections of other institutions such as the Beipiao Pterosaur Museum of China, the Dalian Natural History Museum, and the Chaoyang National Geopark; bringing the total of recovered specimens to over a hundred. Thus, tapejarines seem to represent an important element of the Jehol Biota.

*Sinopterus dongi*, from the Jiufotang Formation (see *Wang & Zhou, 2003a*), was the first tapejarid to be recovered from China, and is the type-species of the genus *Sinopterus*. Subsequently, further six tapejarid species coming from the Jiufotang Fm. have also been proposed: *Sinopterus gui*, *Sinopterus lingyuanensis*, *Huaxiapterus jii*, *Huaxiapterus corollatus*, *Huaxiapterus benxiensis* and *Huaxiapterus atavismus* (see *Wang & Zhou, 2003a*; *Li, Lü & Zhang, 2003*; *Lü & Yuan, 2005*; *Lü et al., 2006a*; *Lü et al., 2006c*; *Lü et al., 2007*; *Lü et al., 2016*). The Jiufotang Fm. tapejarines are involved in a complex series of taxonomic controversies, with the genus *Huaxiapterus* widely considered as a junior synonym of *Sinopterus* (*Wang & Zhou, 2006*; *Wang & Dong, 2008*; *Witton, 2013*; *Zhang et al., 2019*; *Naish, Witton & Martin-Silverstone, 2021*). At the species-level, basically two taxonomic schemes presently exist for the genus *Sinopterus*, which could each be viewed as a conservative and an expansive one. The conservative scheme, proposed by *Witton (2013)* and favored by *Naish, Witton & Martin-Silverstone (2021)*, poses that Jiufotang tapejarines are oversplit, and that most, if not all, Jiufotang tapejarines are members of an ontogenetic continuum of a single species, *Sinopterus dongi*. On the other hand, the expansive scheme of *Zhang et al. (2019)* poses that Jiufotang tapejarines represent at least five valid species within the genus *Sinopterus*: *S. dongi*, *S. corollatus*, *S. benxiensis*, *S. lingyuanensis* and *S. atavismus*. The latter scheme derives from the scheme of *Lü et al. (2016)*, which posed that *Sinopterus* (with three species) and "*Huaxiapterus*" (with four species) are in fact distinct at the generic level, what could thus be considered as a third scheme for the taxonomy of the *Sinopterus* complex (a multi-generic, and also expansive, one).

Despite the abundance of tapejarid remains known from the Jiufotang Formation—and the relevant taxonomic problems involved—not much has been published on the detailed, comparative osteology of *Sinopterus*. Most of the reported specimens were only preliminarily described and figured, to the exception of the recently published IVPP 22388-V, attributed to *Sinopterus atavismus* (*Zhang et al., 2019*). The detailed description of some of these specimens, such as the holotype of *Sinopterus gui*, is further hampered by their poor preservation (*Li, Lü & Zhang, 2003*). This issue becomes particularly problematic due to the complicated taxonomic disputes involved in the *Sinopterus* complex, which cannot be resolved before a detailed reassessment of all known specimens.

Specimen D3072 is a newly reported specimen coming from the Jiufotang Formation, represented by an almost complete, well-preserved postcranial skeleton. Due to its limb proportions and pedal morphology, we attribute it to *Sinopterus dongi*, and defend such attribution under both the conservative or the expansive taxonomic schemes for *Sinopterus*. The main purpose of the present contribution is to offer a detailed account of the osteology of this new specimen, improving knowledge on the anatomy of *Sinopterus dongi*. The importance of such data lies in the need for detailed taxonomic revisions of this genus in the future, which will require a better understanding of the osteology of *Sinopterus*.

## MATERIAL & METHODS

### Specimen and geological setting

Specimen D3072 is permanently stored in the paleontological collection of the Dalian Natural History Museum, in Dalian, Liaoning, China. The specimen was analyzed first-hand and under lenses, measured with the use of calipers, and digitally photographed. Although it lacks precise data on its origin or coordinates, it is catalogued as having been privately collected in the Dapingfang locality (Dapingfang Town, Chaoyang City, Liaoning Province), in rocks belonging to the Jiufotang Formation. The Jiufotang Formation (early Aptian), together with the underlying Yixian Formation (late Barremian—early Aptian), belongs to the Jehol Group. Together (along also with the Barremian Huajiying Formation of the Sichakou-Senjitu Basin), they yield the Jehol Biota, famous for its paleontological richness and exquisite fossil preservation (*e.g.*, *Pan et al., 2013*; *Xu et al., 2020*). The Dapingfang locality has yielded several other Jiufotang Fm. fossils of vertebrate taxa, such as: the azhdarchoid pterosaur *Chaoyangopterus zhangi* (*Wang & Zhou, 2003b*), the istiodactyliform pterosaur *Hongshanopterus lacustris* (*Wang et al., 2008a*; *Wang et al., 2008b*), the istiodactylid *Nurhachius luei* (*Zhou et al., 2019*), the dromaeosaurid *Microraptor gui* (*Xu et al., 2003*), the enanthiornithines *Dapingfangornis sentisorhinus* (*Li et al., 2006*) and *Longipteryx chaoyangensis* (*Zhang et al., 2001*), the non-pygostylian avebrevicaudan *Sapeornis chaoyangensis* (*Zhou & Zhang, 2002a*), the jeholornithiform *Jeholornis prima* (*Zhou & Zhang, 2002b*; *Zhou & Zhang, 2003*), the ornithuromorph *Yanornis martini* (*Zhou et al., 2004*), and the squamate *Yabeinosaurus tenuis* (*Evans, Wang & Li, 2005*).

### Anatomical terminology

In the present study, anatomical terms follow Romerian nomenclature and orientation, using anterior/posterior instead of cranial/caudal (*e.g.*, *Wilson, 2006*). The standardization

of the orientation of wing elements is based on inferred flight position, following *Bennett (2001)*.

## Reference phylogenetic proposal and systematic terminology

The present study follows the reference phylogenetic proposal, and consequent systematic terminology, that tapejarines and thalassodromines are sister-groups (*Kellner & Campos, 2007*; *Vullo et al., 2012*; *Pêgas, Costa & Kellner (2018)*; *Pêgas et al., 2021*). Together, they form the Tapejaridae *sensu Kellner & Campos (2007)*. Under alternative proposals, the Tapejaridae *sensu Lü et al. (2006a)* comprise tapejarines *sensu Kellner & Campos (2007)* but excludes thalassodromines. Tapejarids *sensu Lü et al. (2006a)* are interpreted as a basal group of azhdarchoids, with thalassodromines being more closely related to azhdarchids + chaoyangopterids than to tapejarines –for such alternative proposals, we refer the readers to *Lü et al. (2008)* and *Andres, Clark & Xu (2014)*.

## RESULTS

### Description of new specimen D3072

**Generalities.** The specimen is represented by an almost complete postcranial skeleton preserved on a single slab (Fig. 1, Table 1). The skull, pelvic region and femora are missing. These missing elements (at least the pelvis, sacral vertebrae and femora) might, presumably, have been retained in a counter-slab. Preservation quality varies throughout the skeleton. While some elements are articulated and maintain their natural positions (such as the main wing bones, humerus, radius and ulna, carpals, metacarpal IV and digit IV), some other elements are slightly displaced (especially the scapula and coracoid, but also some small manual and pedal elements).

**Cervical series.** Apart from the atlas and axis, the remaining elements of the cervical series are preserved. All mid-cervicals (cervicals III–VII) are exposed in left lateral view (Fig. 2). These vertebrae are relatively elongate (length/height ratio 1.5–2.5; Table 1), slightly more so than in *Tapejara wellnhoferi* or thalassodromines (length/height ratios of, respectively, ∼2 and 0.9–1.9; see *Vila Nova et al. 2015*; *Leal et al., 2018*), but less so than in chaoyangopterids (ratios 2–3; see *Wu, Zhou & Andres, 2017*; *Leal et al., 2018*) or azhdarchids (ratios 4–10; see *Nessov, 1984*; *Andres, Clark & Xu, 2014*; *Leal et al., 2018*; *Naish & Witton, 2017*; *Pêgas et al., 2021*). Cervical IV is the longest one, as measured from the tips of the prezygapophyses to the tips of the postzygapophyses (Table 1). The preserved cervical vertebrae length formula is III < IV > V > VI > VII > VIII > IX. This formula differs from other azhdarchoid taxa. In *Tapejara wellnhoferi*, cervicals IV and V are of subequal length. According to *Kellner & Hasagawa (1993)*, the same is true for *Tupuxuara leonardii*, but for the sake of precision, we note that in fact cervical V is slightly longer than cervical IV in this taxon (R. Pêgas, pers. obs., 2019). The same is true for chaoyangopterids and azhdarchids, in which cervical V is the longest (*Lü et al., 2008*; *Averianov, 2010*; *Wu, Zhou & Andres, 2017*; *Leal et al., 2018*). There are no mid-cervical ribs.

The neural spine is relatively low, similar to what is seen in *Tapejara* and thalassodromines (*Vila Nova et al. 2015*), which is lower than in pteranodontoids (*e.g., Kellner & Tomida, 2000*; *Bennett, 2001*) but not as low as in chaoyangopterids (*Wu, Zhou*

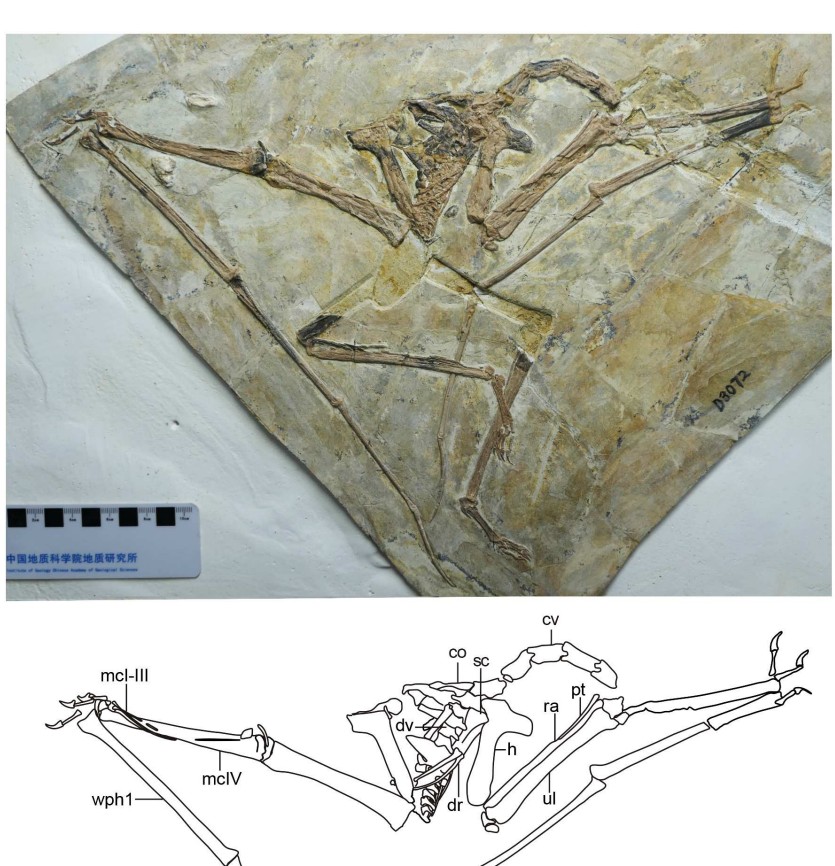

**Figure 1 Specimen D3072, general view.** (A) Photograph and (B) schematic drawing. Abbreviations: cv, cervical vertebrae; co, coracoid; dv, dorsal vertebrae; dr, dorsal rib; fi, fíbula; h, humerus; mc, metacarpal; mt, metatarsal; pt, pteroid; ra, radius; sc, scapula; ti, tibia; ul, ulna; wph, wing phanlanx.

*& Andres, 2017*; *Leal et al., 2018*) nor as in azhdarchids in which the neural spine is only a vestigial ridge (*e.g.*, *Averianov, 2010*).

A single, small pneumatic foramen can be seen on the lateral surfaces of the third, fourth and fifth cervical vertebrae; the status on the sixth and seventh is unclear due to crushing. This foramen is located just below the zygapophyseal ridge, presumably near (ventral to) the contact region between centrum and neural arch. This is similar to the condition seen in *Tapejara wellnhoferi*, in which a single pneumatic foramen on the lateral surface of the mid-cervical vertebrae is present (*Vila Nova et al., 2015*), while two or three

**Table 1 Measurements of the new specimen D3072.** Measurements given in milimeters.

| Elements | Length | Width |
|---|---|---|
| Cervicals 3–7 | 12.4/22.5/19.0/17.4/15.5 | 8.3/9.1/9.6/8.6/10.2 (height) |
| Dorsal series | 66.7 (preserved) | ? |
| Coracoid | 32.8 (preserved) | 5 (shaft) |
| Humerus (l.) | 55.1 | 9.1 (shaft) |
| Deltopectoral crest (l.) | 14.1 | 10.5 |
| Ulna/radius (l.) | 82.6/82.6 | 8.2/? |
| Pteroid (l.) | 32.0 (preserved) | – |
| Metacarpals I–III (r.) | 88.5/34.2/33.3 | ~1 (midshaft) |
| Metacarpal IV (r.) | 90.8 | 8.6 |
| Manual digit 1 (r.), phalanges 1–2 | 11.8/12.5 | ~2 (midshaft)/5.1 (dorsoventral) |
| Manual digit 2 (r.), phalanges 1–3 | ?/14.9/11.3 | ?/~2 (midshaft)/5 (dorsoventral) |
| Manual digit 3 (r.), phalanges 1–4 | 22.0/?/14.5/12.1 | ~1.5 (midshaft)/?/~1.5 (midshaft)/4.6 (dorsoventral) |
| Wing phalanx 1 | 107.7 (r.), 108.6 (l.) | 5.8 (r.), 5.9 (l.) |
| Wing phalanx 2 | 85.6 (r.) | 4.1 (r.), 4.2 (l.) |
| Wing phalanx 3 | 67.8 (r.) | 2.2 (r.), 2.3 (l.) |
| Wing phalanx 4 | 45.0 (r.), 46.5 (l.) | 1.8 (r.), 1.9 (l.) |
| Tibia (r.) | 97.7 | 8.58 (shaft) |
| Metatarsals I-IV | 23.7/23.0/22.6/21.5 (l.) 24.9/23.6/23.1/22.1 (r.) | 1.0/0.8/0.7/1.2 (r.) 1.1/0.9/0.8/1.2 (l.) |

are present in thalassodromines (*Vila Nova et al., 2015*; *Buchmann et al. 2018*) and none in chaoyangopterids and azhdarchids (*Averianov, 2010*; *Wu, Zhou & Andres, 2017*; *Leal et al., 2018*; *Buchmann & Rodrigues, 2019*). Despite the crushed nature of specimen D3072, the openings observed on the lateral surface of these vertebrae match well in size, shape, and location the pneumatic foramina found in the mid-cervical vertebrae of *Tapejara wellnhoferi* (*Eck, Elgin & Frey, 2011*; *Vila Nova et al., 2015*), leading us to interpret these openings in D3072 as pneumatic foramina as well. The tentative identification of a possible lateral pneumatic foramina on the mid-cervicals of *Sinopterus* was already mentioned by *Vila Nova et al. (2015)*, inferred from a depression seen in cervical VII of the holotype of *Sinopterus dongi*. *Zhang et al. (2019)* reported on the presence of this feature in IVPP V 23387, a specimen they referred to *Sinopterus atavismus*. The same condition is seen in the holotype of *Sinopterus lingyuanensis* (*Lü et al., 2016*).

Some details of the mid-cervical vertebrae are obliterated due to crushing. The ventral margins seem to be slightly damaged. In cervical VII, the ventral margin of the centrum is concave, with a relatively large hypapophysis, similar to the condition seen in *Tapejara* (*Vila Nova et al., 2015*). The status in other mid-cervicals is unclear. All mid-cervicals bear well-developed postexapophyses. Cervicals VIII and IX are exposed in a somewhat ventrolateral view and not much can be seen (Fig. 3).

**Dorsal series.** There are 11 preserved dorsal vertebrae (Fig. 3). The anterior margin is concave and the posterior margin is convex. The first three dorsal vertebrae seem to exhibit fused centra and neural arches. Dorsal vertebrae 4 and 5 are not very discernible due to crushing. Although some overlying bones and sediment still obscure most of the centra, a

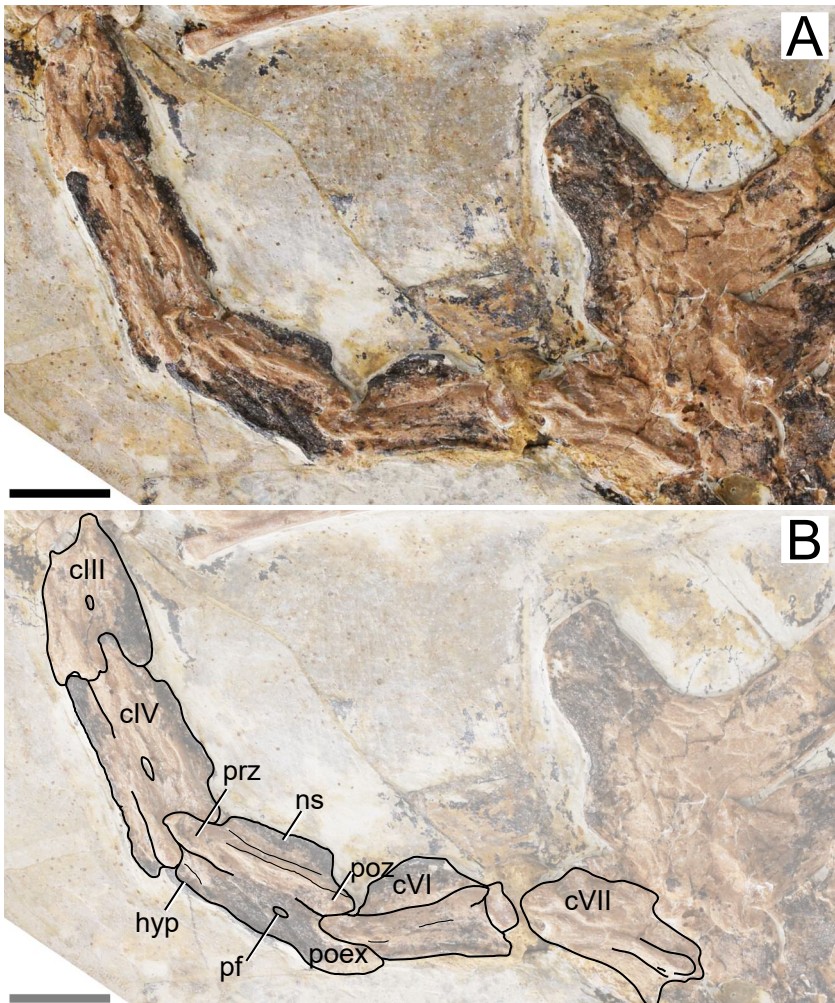

**Figure 2 Specimen D3072, mid-cervical vertebrae.** (A) Photograph and (B) schematic drawing. Abbreviations: c, cervical; hyp, hypapophysis; pf, pneumatic foramen; poex, postexapophysis; poz, postzygapophysis; prz, prezygapophysis. Scale bar equals 10 mm.

contiguous series of transverse processes can be seen, from dorsal vertebrae 1 through 13. A single centrum (probably from dorsal vertebra 6, since is lies ventrally to the sixth dorsal neural arch) is displaced from this contiguous series of neural arches, indicating it was not fused to its respective neural arch. The posterior dorsal vertebrae (7–11) also exhibit an open suture between centra and neural arches. This indicates that, while in mid-cervical vertebrae and anterior dorsal vertebrae the centra were fused to the neural arches, the same was not true for mid- and posterior dorsals. A similar condition was found in the juvenile *Tapejara wellnhoferi* specimen SMNK PAL 1137, in which the preserved cervical vertebrae were entirely fused, while the single preserved dorsal vertebrae lacked fusion between centrum and neural arch (*Eck, Elgin & Frey, 2011*). In dorsal vertebrae 1 and 2, it can be seen that a well-developed fossa is present at the ventral surface of the transverse process base.

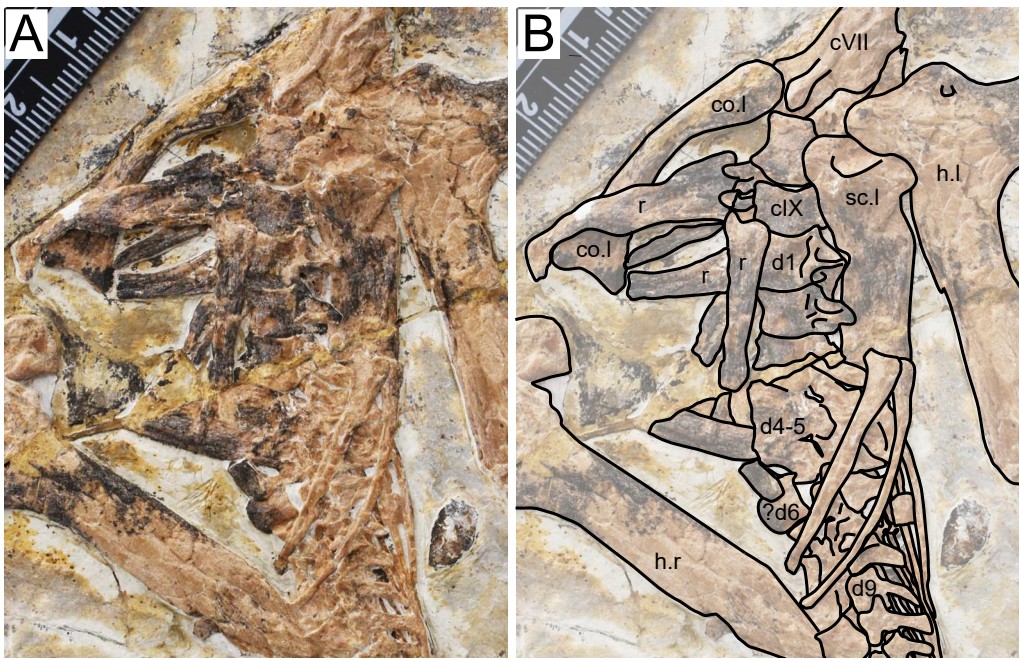

**Figure 3  Specimen D3072, trunk region.** (A) Photograph and (B) schematic drawing. Abbreviations: c, cervical; co, coracoid; d, dorsal; h, humerus; r, rib; sc, scapula.

**Scapula.** The left scapula is preserved medial to the left humerus, in an approximately ventral view (Fig. 3). It is not fused to the coracoid, which is displaced from it. Not much can be observed due to crushing. The shaft of the scapula is wider than that of the coracoid. The scapula is slightly bowed ventrodistally, similarly to *Tapejara wellnhoferi* (*Eck, Elgin & Frey, 2011*), *Caupedactylus ybaka* (*Kellner, 2013*) or *Keresdrakon vilsoni* (*Kellner et al., 2019*).

**Coracoid.** The left coracoid is exposed in dorsal view (Fig. 3). It is curved with an expanded medial end. The medial end is bifid, terminating in an anterior and a posterior eminence, which form the saddle-shaped sternal articulation. These two eminences are approximately equivalent in size, unlike *Tapejara wellnhoferi* (*Eck, Elgin & Frey, 2011*), *Caupedactylus ybaka* (*Kellner, 2013*), *Caiuajara dobruskii* (*Manzig et al., 2014*) or *Keresdrakon vilsoni* (*Kellner et al., 2019*) in which the posterior eminence is slightly larger than the anterior one.

**Humerus.** Both humeri are preserved approximately in ventral view (Fig. 4). The left humerus is preserved in a better condition than the right one, in which the ulnar crest, deltopectoral crest and distal region are slightly damaged. The shaft of the humerus is straight. The deltopectoral crest is subrectangular in profile and is located proximally. The ventral margin is straight. It forms an approximately perpendicular angle relative to the main shaft. This is similar to *Caupedactylus ybaka* (*Kellner, 2013*) and *Tupuxuara leonardii* (*Witton, Martill & Green, 2009*), and slightly different from *Tapejara wellnhoferi*, *Caiuajara dobruskii* and *Tupandactylus navigans*, in which the deltopectoral crest is at a slightly oblique angle to the humeral shaft, ventromedially oriented (*Eck, Elgin & Frey, 2011;*

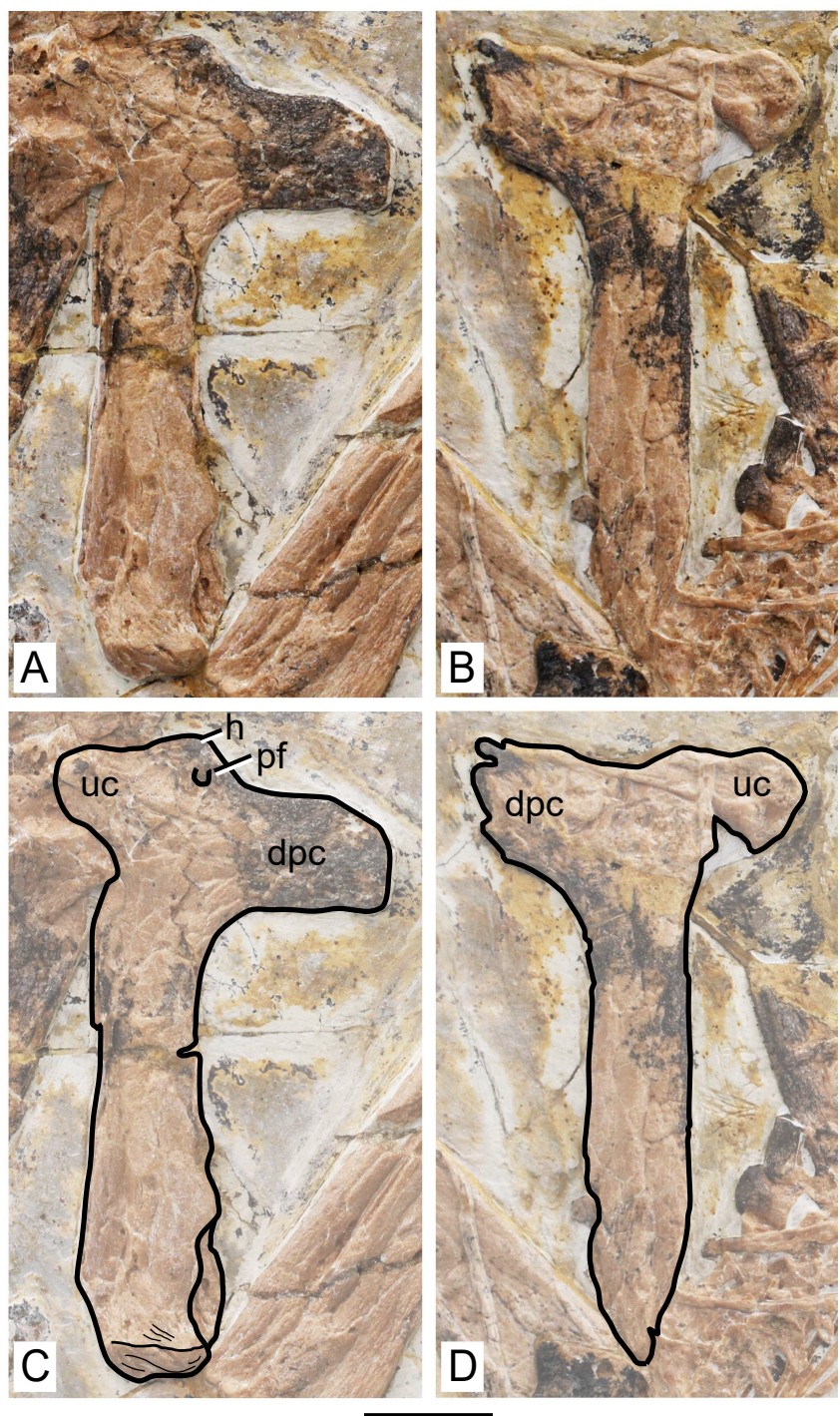

**Figure 4** **Specimen D3072, humerus.** Photograph of (A) left and (B) right humeri. Schematic drawing of (A) left and (B) right humeri. Abbreviations: dpc, deltopectoral crest; h, head; pf, pneumatic foramen; uc, ulnar crest. Scale bar equals 10 mm.

*Manzig et al., 2014*; *Beccari et al., 2021*). The region of the humeral head, in ventral view, is marked by a triangular eminence. The ulnar crest is rounded and posterodorsally flared (so that it is well visible in ventral view), similarly to *Tapejara wellnhoferi* and *Caiuajara dobruskii* (*Eck, Elgin & Frey, 2011*; *Manzig et al., 2014*). A distinct foramen is present on the ventral proximal end of the humerus, close (slightly distal to) the humeral head, between the deltopectoral and ulnar crests. This is similar to other azhdarchoids in general, as can be seen in *Tapejara wellnhoferi* (*Eck, Elgin & Frey, 2011*), *Caiuajara dobruskii* (*Manzig et al., 2014*), *Tupandactylus navigans* (*Beccari et al., 2021*), *Caupedactylus ybaka* (*Kellner, 2013*), *Tupuxuara leonardii* (*Kellner & Hasegawa, 1993*), *Keresdrakon vilsoni* (*Kellner et al., 2019*), and azhdarchids (*Lawson, 1975*; *Hone, Habib & Therrien, 2019*). The humeral shaft slightly expands towards the distal end. Two humeral epiphyses are present on the left side. They are unfused to the humerus and preserved near the proximal region of the ulna. Since they are displaced from the humerus, their relative orientation is unclear.

**Radius.** The ulna and radius are preserved parallel to each other on both sides. The diameter of the ulna is larger than that of the radius. A precise ratio cannot be given, since the ulna partially overlaps the radius on both sides, but the radius seems to be, roughly, not less than half the diameter of the ulna. Not much can be observed on the right side, since the ulna and radius are too compressed against each other, and the proximal and distal regions are slightly damaged. The left side can be better observed, although crushing still obscures some details. The left radius and ulna are exposed in an approximately anteroventral view.

The proximal region of the left radius bears a concave proximal margin (Fig. 5). The proximal surface itself bears a cotyle. Two eminences can be seen, an anterodorsal and a posteroventral one. The anterodorsal one (the biccipital process) is smaller than the posteroventral one and projects only gently anterodorsally, similarly to *Tapejara wellnhoferi* (*Eck, Elgin & Frey, 2011*) and *Jidapterus edentus* (*Wu, Zhou & Andres, 2017*). The posteroventral one, which is larger, projects proximally, more so than in *Tapejara wellnhoferi* (*Eck, Elgin & Frey, 2011*) or *Jidapterus edentus* (*Wu, Zhou & Andres, 2017*) in which it is less projected, or in *Pteranodon* in which it is rather inconspicuous (*Bennett, 2001*). The distal end of the radius is slightly expanded, but not much can be seen due to crushing.

**Ulna.** The left ulna is better preserved than the right one, as mentioned above. The shaft is straight. On the proximal end, a dorsal, large eminence projects proximally, presumably for the attachment of M. *triceps brachii* (Fig. 5). This eminence is particularly prominent, more so than in *Tapejara wellnhoferi* (*Eck, Elgin & Frey, 2011*), *Jidapterus edentus* (*Wu, Zhou & Andres, 2017*) or *Pteranodon* (*Bennett, 2001*). Ventral to this eminence, the capitular and trochlear cotyles can be seen, separated by a discrete ridge. They both face proximally. Their shapes and dimensions cannot be assessed confidently, since the region has been very compressed. On the distal end, two collateral processes can be seen, a dorsal and a ventral one (Fig. 5). The dorsal one is much less developed than the ventral one, similarly to *Tapejara wellnhoferi* (*Eck, Elgin & Frey, 2011*) and the possible tapejarid "*Santanadactylus spixii*" (*Wellnhofer, 1987*; *Kellner & Tomida, 2000*) and unlike *Pteranodon* in which they are roughly subequal (*Bennett, 2001*). The dorsal articular surface, in the form of a gentle depression, can be seen on the distal surface of the dorsal collateral process. The olecranon
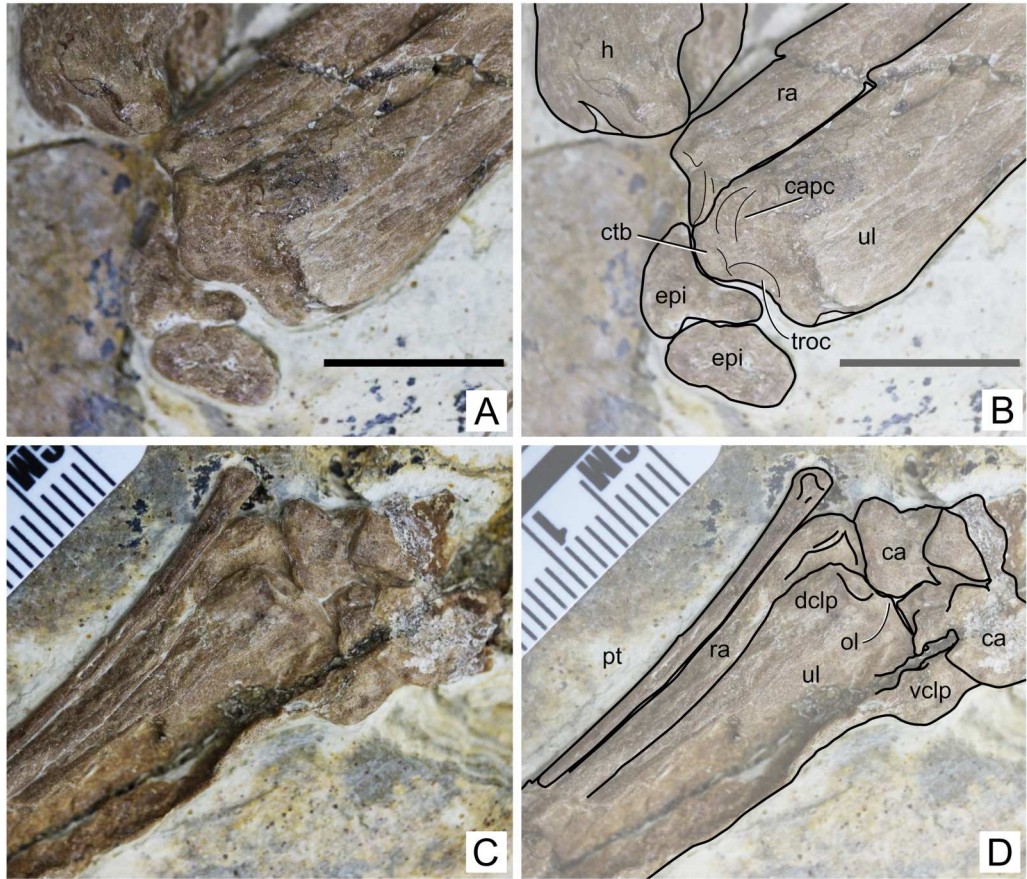

**Figure 5** **Specimen D3072, radius and ulna.** Proximal region, (A) photograph and (B) schematic drawing. Scale bar equals 10 mm. Distal region, (A) photograph and (B) schematic drawing. Abbreviations: ca, carpal; capc, capitular cotyle; ctb, crest for M.*triceps brachii*; dclp, dorsal collateral process; epi, epiphysis; h, humerus; ol, olecranon; pt, pteroid; ra, radius; troc, trochlear cotyle; ul, ulna; vclp, ventral colateral process.

process is only slightly prominent, similar to *Pteranodon* (*Bennett, 2001*) and unlike "*Santanadactylus spixii*" and *Tapejara wellnhoferi* in which it is larger and more prominent (*Wellnhofer, 1987*; *Eck, Elgin & Frey, 2011*). Further details on the proximal surface are unclear due to crushing.

**Carpals.** Both left and right carpal regions are preserved. On the left region, the bones are less crushed, and it is clear that all carpal elements are unfused, although not much information can be retrieved (Fig. 5). On the right side, the elements are very compressed against each other and their limits thus became unclear, except for the preaxial carpal and pteroid which are slightly displaced (Fig. 6). The pteroid can be seen on both sides. On the left side, it is almost complete, lacking only the very apex tip. It is exposed in anterior view. It is thin, rod-like and elongated. The shaft is gently bowed, and the base is slightly expanded. On the right side, only a small fragment of the bone is preserved, also not in natural position.

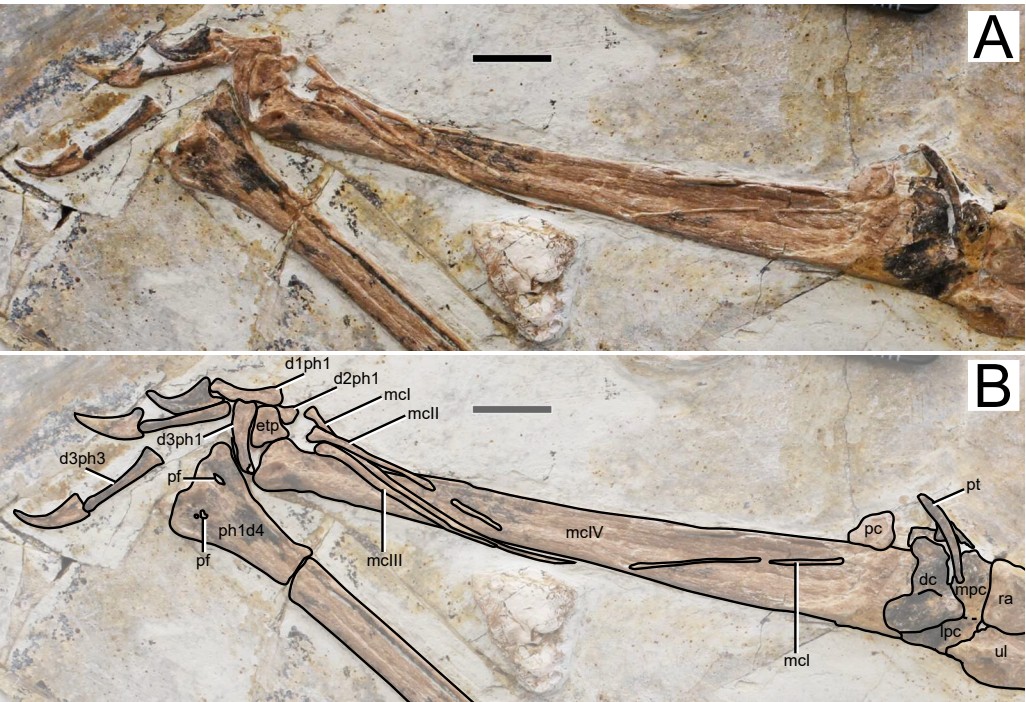

**Figure 6 Specimen D3072, right carpus and manus.** (A) Photograph and (B) schematic drawing. Abbreviations: d, digit; dc, distal carpals; etp, extensor tendon process; lpc, lateral proximal carpal; mc, metacarpal; mpc, medial proximal carpal; pt, pteroid; ra, radius; ul, ulna; pc, preaxial carpal; pf, pneumatic foramen; ph, phanlanx; pt, pteroid. Scale bar equals 10 mm.

**Metacarpals.** Metacarpals I–III are slender, thin, rod-like bones, while metacarpal IV is robust as in other pterosaurs. Metacarpal I is not entirely preserved, as attested by a missing mid-shaft portion (Fig. 6). Still, it extends for about 90% the distance between its distal tip and the carpal region. The proximal tip seems to be damaged and it is likely that it extended further proximally, possibly onto the carpal region, contacting it, as has been reported for the holotypes of *Sinopterus dongi* and "*Huaxiapterus*" *jii* (*Wang et al. 2003*; *Lü & Yuan, 2005*). Metacarpals II and III are comparatively shorter, extending for about a third of metacarpal IV. They are pointed proximally, and slightly expanded distally. Metacarpal IV is wider proximally than distally. The distal region exhibits the typical trochlear-like morphology that is seen in pterosaurs, with a pair of condyles, a dorsal one and a ventral one (*e.g., Wellnhofer, 1978*). Not much details can be seen on the condyles due to crushing.

**Free digits.** On the right side, the manus is almost entirely preserved, except for the second phalanx of digit 3 (Fig. 6). On the left side, only two phalanges are present for each digit, the two phalanges of digit one and the two distal ones of digits 2 and 3 (Fig. 1). The digits become progressively slightly longer from 1 to 3. The proximal phalanx of right digit 1 has a well-developed trochlear-like articulation with two small condyles on the proximal tip. The condition in the other proximal phalanges is unclear. The first digit bears only two phalanges (the proximal phanalx and the ungual), as typical of pterosaurs (*e.g., Wellnhofer, 1978*). The second digit bears three phalanges. On the third digit, only three phalanges are

preserved, although the typical condition in pterosaurs is the presence of four phalanges (*e.g.*, *Wellnhofer, 1978*). The proximal phalanx is disarticulated, displaced from its natural position. The two distalmost phalanges are articulated to each other, from a distance from the proximal element. It is plausible that the third phalanx of digit 3, which is typically small and square-like, has been displaced and possibly obscured by the other elements. The manual claws are long and recurved. They are slightly larger than the pedal claws, with relatively larger flexor tubercles as well.

**Wing digit.** Wing phalanges I–IV decrease in length progressively from I to IV (Fig. 1). Phalanges 1–3 are straight. On the ventral surface of the proximal end of the right first phalanx, two pneumatic openings are present, one on the posterior region and another on the anterior region. Such feature has only ever been reported for the azhdarchoid *Keresdrakon vilsoni* (see *Kellner et al., 2019*). The fourth wing phalanx (as seen on both left and right sides) is distinctively curved, as seen in *Eopteranodon lii* (*Lü & Zhang, 2005*; *Lü et al., 2006b*) and the holotypes of *Sinopterus dongi* (*Wang & Zhou, 2003a*), *S. jii* (*Lü & Yuan, 2005*) and *"H." atavismus* (*Lü et al., 2016*), but unlike *"Huaxiapterus" corollatus* in which it is straighter (*Lü et al., 2006a*).

**Tibia.** The right tibia is highly damaged, with displaced splints of crushed bone along its shaft. The left tibia lacks the proximal region. As can be seen on both sides, neither the fibula nor the proximal tarsals are fused to the tibia (Fig. 7). The tibia is broader proximally than distally.

**Fibula.** On both sides, it can be seen that the fibula is not fused to the tibia. The fibula is present as a very thin, elongate bone, parallel to the tibia. It runs through, approximately, a third the length of the tibia.

**Tarsus.** The tarsus can be observed on both sides (Fig. 7). Two proximal and two distal elements are present on the left pes, all slightly displaced from their presumed natural positions. On the right pes, the lateral proximal tarsal and both distal tarsals are preserved. The medial proximal tarsal is not preserved. A small, rounded, unidentified bone is present. The proximal tarsals are not fused to the tibia. On both sides, the lateral proximal tarsal is approximately rectangular and is the largest tarsal element, as in IVPP V 23388-V (*Zhang et al., 2019*).

**Metatarsus.** Metatarsals I-IV are elongate, slender bones, slightly expanded at the ends (Fig. 7). The proximal ends are broader than the distal ones, particularly in metatarsal IV. They are progressively shorter from I to IV, as in the holotype of *Sinopterus dongi* (*Wang & Zhou, 2003a*) and unlike other purported species of the *Sinopterus* complex (*Zhang et al., 2019*). Metatarsal V is reduced and hook-shaped, with a broad proximal region and a pointed distal apex. It articulates with the lateral distal tarsal.

**Pedal digits.** The pedal formula can be inferred to be 2-3-4-5-1 (Fig. 7). The length ratio of metatarsal III to tibia is 0.24. The pedal claws and the manual claws have roughly the same shape, except that the manual claws are slightly larger and bear slightly larger flexor tubercles. A keratinous ungual sheath is preserved on right pedal digit 4 (Fig. 7). The fifth digit is represented by a single, extremely reduced phalanx, which is preserved on the left side.

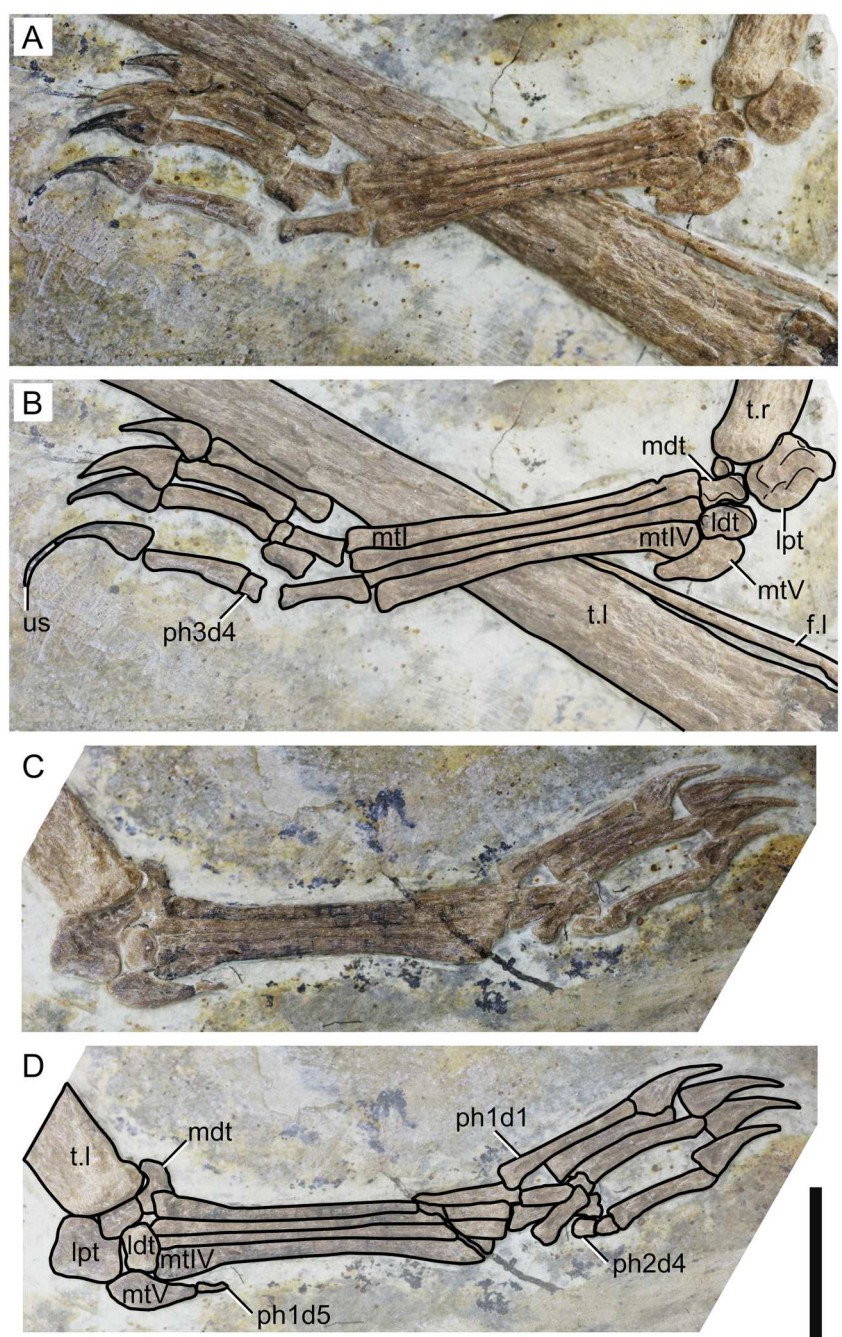

**Figure 7 Specimen D3072, pedes.** Right pes, (A) photograph and (B) schematic drawing. Left pes, (C) photograph and (D) schematic drawing. Abbreviations: f, fibula; ldt, lateral distal tarsal; lpt, lateral proximal tarsal; mdt, medial distal tarsal; mt, metatarsal; t, tibia; us, ungual sheath. Scale bar equals 10 mm.

## DISCUSSION

### Identification of specimen D3072

Based on the relatively low level of postcranial bone fusion, it is clear that specimen D3072 is a juvenile. The new specimen lacks fusion of: the humeral epiphyses, scapulocoracoid, the extensor tendon process of the first wing phalanx, the carpal elements, tibia and fibula, tibia and proximal tarsals, and neural arches and centra of most dorsal vertebrae. The only postcranial elements that are fused are the neural arches and centra of cervical vertebrae and anterior dorsal vertebrae. Despite the uncertainties surrounding studies on the ontogeny of pterosaurs (see *Kellner, 2015*; *Dalla Vecchia, 2018*) and reptiles overall (*Griffin et al. 2021*), such low level of skeletal fusion is indicative of a juvenile nature for this specimen (*e.g.*, *Bennett, 1993*; *Kellner, 2015*; *Dalla Vecchia, 2018*; *Griffin et al. 2021*). With a wingspan of about 1,134 mm, D3072 is similar in size to holotype of *Sinopterus dongi*, which is also a juvenile, with 1,200 mm in wingspan (*Wang & Zhou, 2003a*), and slightly larger than other juveniles such as the holotypes of *S. gui* (with ∼800 mm in wingspan; *Li, Lü & Zhang, 2003*) and *"H." atavismus* (∼850 mm in wingspan; *Lü et al., 2016*). These are substantially larger than the possible near-hatchling *Sinopterus* specimen represented by the holotype of *Nemicolopterus crypticus* (*Witton, 2013*; *Naish, Witton & Martin-Silverstone, 2021*), with a wingspan of ∼250 mm. In contrast, D3072 is considerably smaller than subadult specimens referred to the genus *Sinopterus* such as BXGM V0011 and IVPP V 23388, both with 1,600 mm in wingspan (*Lü et al., 2007*; *Zhang et al., 2019*), or the adult specimen D2525, with 2,000 mm in wingspan (*Lü et al., 2006c*).

The new specimen D3072 is flattened, preserved on a slab, as typical of Jehol fossils (*e.g.* *Pan et al., 2013*; *Xu et al., 2020*) and many other pterosaur remains from around the world (*e.g.*, *Beccari et al., 2021*). This frequent preservation style in pterosaurs turns anatomical comparisons relatively hard, since it limits what bone surfaces can, or not, be viewed (*e.g.*, in D3072, the anterior and posterior views of the cervical vertebrae cannot be seen, nor the distal surface of the humerus). Despite this obstacle, there are a number of observable features in specimen D3072 that allow its identification. It exhibits a subrectangular, non-warped deltopectoral crest of the humerus, typical of the Tapejaroidea *sensu Kellner (2003)*. Specimen D3072 further exhibits features that are found in tapejarines such as mid-cervical vertebrae comparatively short with low (but not ridge-like) neural spines and a single pneumatic foramen on the lateral surface (*Vila Nova et al., 2015*), as well as a dorsally flared ulnar crest of the humerus similar to *Tapejara* and *Caiuajara* (*Eck, Elgin & Frey, 2011*; *Manzig et al., 2014*). Together, all of these features are found only in tapejarines such as *Tapejara* (*e.g.*, *Eck, Elgin & Frey, 2011*; *Vila Nova et al., 2015*), *Caiuajara* (*Manzig et al., 2014*) and *Sinopterus* (*Wang & Zhou, 2003a*; *Lü et al., 2016*; *Zhang et al., 2019*). Limb proportions also match the typical condition seen in tapejarines, including *Sinopterus dongi* (see *Vila Nova & Sayão, 2012*).

The fact that specimen D3072 is a tapejarine coming from the Jiufotang Formation indicates that it likely belongs within the *Sinopterus* complex. The next question is whether it matches, or not, any of the proposed species of *Sinopterus*. Under the expansive scheme defended by *Zhang et al. (2019)*, there is reason to regard D3072 as a specimen of *Sinopterus*

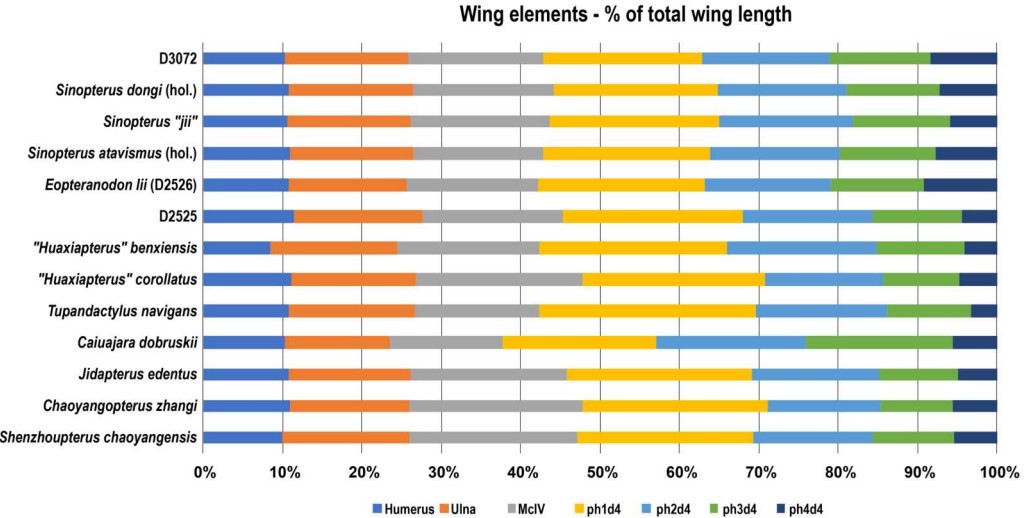

**Figure 8** **Wing elements proportions (in percentage of total wing length) in selected azhdarchoids.**
Data source: *Sinopterus dongi* (holotype, IVPP V 13363; *Wang & Zhou (2003a)*; *Sinopterus "jii"* (holotype,
GMN-03-11-001; *Lü & Yuan, (2005)*; *S.* atavismus (holotype, XHPM 1009; *Lü et al., 2016)*; *Eopteranodon
lii* (D2526, *Lü et al., 2006b)*; D2525 (*Lü et al., 2006c)*; *"Huaxiapterus" benxiensis* (holotype, BXGM V0011;
*Lü et al., 2007)*; *"Huaxiapterus" corollatus* (holotype, ZMNH M8131), *Lü et al. (2006a)*; *Tupandactylus
navigans* (GP/2E 9266), Beccari et al., (2021); *Caiuajara dobruskii* (composite), *Manzig et al. (2014)*;
*Jidapterus edentus* (holotype, RCPS-030366CY), *Wu, Zhou & Andres (2017)*; *Chaoyangopterus zhangi*
(holotype, IVPP V 13397), *Wang & Zhou (2003b)*; *Shenzhoupterus chaoyangensis* (holotype, HGM
41HIII-305A), *Lü et al. (2008)*. Abbreviations: d, digit; hol., holotype; McIV, metacarpal IV; ph, phalanx.

*dongi*: this new specimen exhibits a metatarsal I longer than metatarsals II–IV, which are
progressively shorter. This feature is absent in other proposed species of the genus, and is
suggested as a diagnostic feature of *Sinopterus dongi* under the expansive scheme (*Zhang
et al., 2019*). Furthermore, regarding limb proportions, specimen D3702 closely matches
the holotype of *Sinopterus dongi*, (*Wang & Zhou, 2003a*; *Wang & Zhou, 2003b*; *Vila Nova
& Sayão, 2012*; see also Fig. 8). Under the conservative scheme proposed by *Witton (2013)*
and preliminarily defended by *Naish, Witton & Martin-Silverstone (2021)*, there is also no
reason to regard D3072 as distinct from the type-species *Sinopterus dongi*, with which
it shares pedal morphology and similar limb proportions. Thus, under either scheme,
specimen D3072 can confidently be attributed to the species *Sinopterus dongi*. How many
further *Sinopterus* species are valid or not, as well as their respective diagnostic features, are
matters to be explored elsewhere, pending detailed accounts of the comparative osteology
of further specimens.

## Comments on the osteology of *Sinopterus dongi* and implications

The new specimen D3072 sheds fresh light on the osteology of *Sinopterus dongi*,
corroborating previous suggestions and revealing new data. For instance, *Vila Nova
et al. (2015)* were the first authors to suggest that pneumatic foramina were most likely
present on the lateral surface of the centrum in *Sinopterus*, which were previously regarded
as absent (*Lü et al., 2006b*; *Liu et al., 2015*). Later, *Zhang et al. (2019)* reported on the

presence of this feature for IVPP V 23388 (therein referred to *S. atavismus*), but refrained from attesting its presence in other species of *Sinopterus*. In D3072, this feature is clearly preserved, suggesting that it was most likely common for all potential *Sinopterus* species. Specimens where it cannot be seen are probably simply affected by taphonomy, such as the holotype of *S. dongi*, as suggested by *Vila Nova et al. (2015)*.

A purportedly distinctive scapula, described as strongly curved, has been considered as a diagnostic feature of *Sinopterus* (*Wang & Zhou, 2003a*; *Zhang et al., 2019*), although it has not been demonstrated how such strong curvature differs from what is seen in other pterosaurs. If compared to other tapejarines, the curvature seen in the scapula of *Sinopterus dongi* (as seen from anterior or posterior views) is comparable to what is seen in *Tapejara wellnhoferi* (*Eck, Elgin & Frey, 2011*), *Caupedactylus ybaka* (*Kellner, 2013*) and *Caiuajara dobruskii* (*Manzig et al., 2014*). If compared to other azhdarchoids, it is also comparable to the chaoyangopterid *Jidapterus edentus* (*Wu, Zhou & Andres, 2017*).

The coracoid, in turn, does seem to be distinctive in *Sinopterus dongi* if compared to other tapejarines. The coracoid of D3072 is notably bowed in dorsal view, more so than in *Tapejara*, *Caupedactylus* or *Caiuajara* (*Eck, Elgin & Frey, 2011*; *Kellner, 2013*; *Manzig et al., 2014*). The proximal pair of eminences also exhibit a distinguishing condition, being similar in size, unlike *Tapejara*, *Caupedactylus* or *Caiuajara* in which posterior eminence is slightly larger than the anterior one. The condition in further proposed species of the *Sinopterus* complex remains to be described. This feature is thus of interest for future comparative studies involving Jiufotang tapejarines.

The overall anatomy of the humerus agrees well with *Tapejara* (see *Eck, Elgin & Frey, 2011*) and *Caiuajara* (see *Manzig et al., 2014*), due to the presence of an approximately straight shaft, a rectangular deltopectoral crest, a humeral head eminence that is triangular in ventral view, and a rounded, dorsally flared ulnar crest. This last character is absent in other azhdarchoids such as *Caupedactylus* (*Kellner, 2013*), *Tupuxuara* (*Witton, Martill & Green, 2009*) and azhdarchids (*Witton, Martill & Green, 2009*) in which the ulnar crest is trapezoidal and posteriorly oriented, and this feature is thus of potential phylogenetic relevance for tapejarines. Furthermore, *Sinopterus dongi* differs from *Tapejara*, *Caiuajara* and *Tupandactylus* in exhibiting a deltopectoral crest of the humerus which is perpendicular to the humeral shaft, while in *Tapejara*, *Caiuajara* and *Tupandactylus* it is slightly oblique, slanting proximally (*Eck, Elgin & Frey, 2011*; *Manzig et al., 2014*; *Beccari et al., 2021*; see also Fig. 9). Future comparative and phylogenetic studies should take these features into account.

In the new specimen D3072, metacarpal I is very elongate and, as mentioned above, probably contacted the carpus, as in the holotypes of *Sinopterus dongi* and "*Huaxiapterus*" *jii* (*Wang et al. 2003*; *Lü & Yuan, 2005*). This feature is worthy of further investigation, since a short metacarpal I (extending for only a third of the length of metacarpal IV) is a potential distinguishing feature between "*Huaxiapterus*" (as in the holotypes of "*H.*" *benxiensis* and "*H.*" *corollatus*) and other *Sinopterus* species (*Kellner & Campos, 2007*). Future assessments of the holotypes of "*H.*" *benxiensis* and "*H.*" *corollatus* are needed in order to verify this feature, by investigating whether it is a natural feature or a potential preservation artifact.

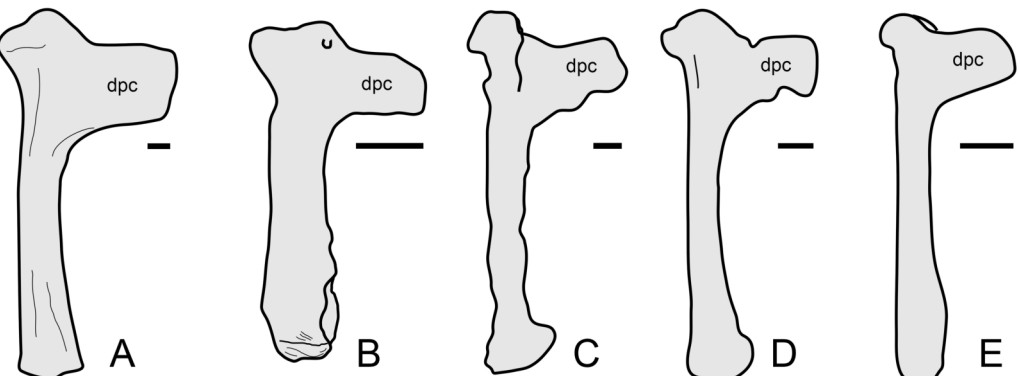

**Figure 9** **Humerus morphology in tapejarines.** (A) *Caupedactylus ybaka*, right humerus in anterior view, drawn from *Kellner (2013)*. (B) D3072, left humerus in an approximately ventral view. (C) *Tupandactylus navigans*, mirrored right humerus in posterior view, drawn from Beccari et al., (2021). (D) *Caiuajara dobruskii*, mirrored left humerus in anterior view, drawn from *Manzig et al. (2014)*. (E) *Tapejara wellnhoferi*, right humerus in anterior view, drawn from *Eck, Elgin & Frey (2011)*. Notice the variation in the orientation of the deltopectoral crest, perpendicular to humeral shaft in (A) and (B), and oblique in (C), (D) and (E). Scale bars equal 10 mm.

Interestingly, the proximal region of the first wing phalanx of D3072 presents two ventral pneumatic foramina, an anterior and a posterior one. This differs from other tapejarines such as *Tapejara wellnhoferi* (*Eck, Elgin & Frey, 2011*), *Caupedactylus ybaka* (*Kellner, 2013*) and *Caiuajara dobruskii* (*Manzig et al., 2014*). It also differs from the thalassodromine *Tupuxuara leonardii* (R. Pêgas, pers. obs., 2019) and the chaoyangopterid *Jidapterus edentus* (*Wu, Zhou & Andres, 2017*), in which a single foramen is present. A pair of foramina on the ventral side of the first wing phalanax proximal region has been reported only for the azhdarchoid *Keresdrakon* (*Kellner et al., 2019*). Because of this peculiarity, it is worth investigating how widespread is this feature within Jiufotang tapejarines, what may prove useful in future revisions of the *Sinopterus* complex.

Another interesting feature observable in the new specimen D3072 is the ungual sheath preserved on the right pedal digit 4 (Fig. 7). The sheath extends for about 50 mm, which is about 85% the length of the bony ungual. Keratinous sheaths in tapejarid unguals were previously known only from an incomplete skeleton from the Crato Formation, SMNK PAL 3830, which also exhibits an elongate and well-recurved ungual sheath (*Frey et al., 2003*). Specimen D3072 represents the first case of ungual sheath preservation in a Chinese tapejarid. Tapejarine tapejarids such as *Sinopterus* specimens from the Jiufotang Formation, as well as SMNK PAL 3830, are known to exhibit ungual phalanges that are particularly robust and recurved if compared to other pterosaurs such as pteranodontoids, azhdarchids and chaoyangopterids (*Wu, Zhou & Andres, 2017*). The robust, recurved unguals of tapejarines have been interpreted as potentially related to scansorial or arboreal habits, by favoring climbing of tree trunks (*Wang & Zhou, 2003a*). This interpretation was corroborated by morphometric analyses by *Wu, Zhou & Andres (2017)*, which revealed that, while chaoyangopterid unguals are consistent with those of ground-dwelling tetrapods, those of tapejarines were consistent with those of tetrapods with specialized claws for

arboreal, scansorial, or predatory lifestyles (*Wu, Zhou & Andres, 2017*). The new specimen D3072 shows that, in life, the pedal claws of *Sinopterus* were substantially longer and more recurved than suggested by bone alone (as in SMNK PAL 3830), due to a significantly elongate horny sheath. This further highlights the specialized morphology of tapejarine unguals, possibly related to arboreal or scansorial habits (*Wu, Zhou & Andres, 2017*).

## Short comments on the *Sinopterus* complex

Albeit relatively recent, the taxonomic history of the genus *Sinopterus* is convoluted. Since the description of *Sinopterus dongi* (*Wang & Zhou, 2003a*), further 6 species were named until 2016 (see *Lü et al., 2016*). *Witton (2013)* raised the concern that all Jiufotang tapejarines could represent an ontogenetic continuum of a single species (including also *Nemicolopterus crypticus*, interpreted as a potential hatchling specimen). Later on, further two new species were proposed: *Sinopterus lingyuanensis* and "*Huaxiapterus*" *atavismus* (*Lü et al., 2016*), which were later regarded as valid (both under the genus *Sinopterus*) by *Zhang et al. (2019)*, who recognized at least five valid species for *Sinopterus*. More recently, *Naish, Witton & Martin-Silverstone (2021)* corroborated the proposition of *Witton (2013)* that most, if not all, proposed species could be conspecific (including the most recently named ones, *Sinopterus lingyuanensis* and "*Huaxiapterus*" *atavismus*). These authors demonstrated that all Jiufotang tapejarines seemed to fall within a single spectrum of variation, as demonstrated by trends in limb and crest proportions (*Naish, Witton & Martin-Silverstone, 2021*). However, they expressed caution regarding this proposition. *Naish, Witton & Martin-Silverstone (2021)* noticed that the holotype of "*Huaxiapterus*" *corollatus* seems to be an outlier regarding limb proportions and, thus, could represent a second taxon. These authors further reiterated that, before the taxonomy of the *Sinopterus* complex can be resolved, a detailed anatomical reassessment of the known specimens is needed (*Naish, Witton & Martin-Silverstone, 2021*).

In the present work, we concur with the observations and preliminary conclusions provided by *Naish, Witton & Martin-Silverstone (2021)* regarding these issues. We regard that there is strong evidence for a taxonomic oversplitting in the *Sinopterus* complex, although it is still possible that more than one taxon is present in the known sample of Jiufotang tapejarines (see *Naish, Witton & Martin-Silverstone, 2021*). In order to try to untangle the *Sinopterus* complex, a detailed comparative reassessment of all known specimens is thus paramount. However, before known specimens can be compared, detailed accounts of their osteology need to be provided. So far, few works have focused on the detailed osteological description of *Sinopterus* specimens (*e.g.*, *Zhang et al., 2019*). Detailed osteological redescriptions are thus much needed for most of the known specimens (such as the types of *Sinopterus gui*, *Sinopterus lingyuanensis* and *Huaxiapterus atavismus*). Given the importance of osteological descriptions as foundations for subsequent taxonomic decisions and systematic studies, in the present contribution we aimed at providing osteological data for the new specimen D3072.

Here, we corroborate the presence of several features in *Sinopterus dongi*: pneumatic foramina piercing the lateral surface of mid-cervical vertebrae, a metacarpal I reaching the carpus, and a metatarsal I longer than subsequent metatarsals. Based on the configuration

of metacarpal I, it does seem plausible that multiple species of *Sinopterus* could be present in the Jiufotang assemblage, and not a single one. Further examination of the already known specimens, with detailed descriptions, are needed in order to confirm whether the variation in metacarpal I is natural or, perhaps, preservational. Based on metatarsal proportions, it does also seem plausible that not all *Sinopterus* specimens are conspecific, and that, instead, more than one species is present. Still, detailed accounts of these variations still need to be performed taking into account the entire sample of known specimens.

From this point on, further anatomical accounts concerning the other proposed species must be made. With time, this should allow a better understanding of osteological variation in the *Sinopterus* complex and, subsequently, lead to a revised taxonomic scheme for the genus. Hopefully this study will serve as a comparative basis for future works focusing on the osteology and taxonomy of the *Sinopterus* complex.

## CONCLUSIONS

The new specimen D3072 represents a postcranial skeleton of a Jiufotang tapejarine. Under both the conservative and expansive taxonomic approaches that presently exist regarding the *Sinopterus* complex, specimen D3072 is attributable to *Sinopterus dongi*, with which it shares almost identical limb proportions and pedal morphology, with metatarsal I being the longest. The osteological description we provide for this new specimen sheds new light on the anatomy of *Sinopterus*, which should be helpful for future systematic studies as well as a taxonomic revision of the *Sinopterus* complex.

In the present study, we confirm the presence of pneumatic foramina on the lateral surface of mid-cervical vertebrae in *Sinopterus dongi*. We further report that this taxon distinguishes itself from *Tapejara wellnhoferi* (the next best-known tapejarine so far) in that cervical IV is the longest one, the two proximal coracoid eminences are similar in size, the deltopectoral crest of the humerus is perpendicular to the humeral shaft, and that two ventral pneumatic foramina are present on the proximal region of wing digit 1. We also note that *Sinopterus dongi* shares with *Tapejara wellnhoferi*, as well as *Caiuajara dobruskii*, some features that are absent outside of the Tapejarinae within azhdarchoids, such as a rounded and dorsally flared ulnar crest of the humerus, a single and small pneumatic foramen on the mid-cervical vertebrae, and a well-developed hypapophysis. We further report for the first time an ungual sheath in a pedal ungual of *Sinopterus dongi*. It demonstrates that these structures were well-developed in this taxon, further enhancing the elongation and the curvature of the pedal claw, similar to the indeterminate Crato tapejarid SMNK PAL 3830.

The conditions of all of these features remain to be assessed in other purported species of the *Sinopterus* complex, as well as on other tapejarids that were only preliminarily described such as *Caiuajara dobruskii* (*Manzig et al., 2014*) and *Tupuxuara leonardii* (Kellner & Hasegawa, 1993), and also taxa for which postcranial skeletons are still unknown (such as *Tupandactylus* spp. or *Europejara olcadesorum*). Accounts on the osteology of these other forms will be needed in order to assess potential diagnostic and phylogenetic signals in these features, which is possibly a promising issue to explore. We hope the osteological data presented here will serve as basis for comparative studies exploring further tapejarines overall, and especially those from the Jiufotang Formation.

**Institutional Abbreviations**

| | |
|---|---|
| **BMNH (=BPV)** | Beijing Museum of Natural History, Beijing, China |
| **BXGM** | Benxi Geological Museum, Benxi, China |
| **D** | Dalian Natural History Museum, Dalian, China |
| **GMN** | Geological Museum of Nanjing, Nanjing, China |
| **IVPP** | Institute of Vertebrate Paleontology and Paleoanthropology, CAS, Beijing, China |
| **JPM (=JZMP)** | Jinzhou Paleontological Museum, Jinzhou, China |
| **PMOL (=LPM)** | Paleontological Museum of Liaoning, Shenyang, China |
| **XHPM** | Xinghai Paleontological Museum, Dalian, China |
| **ZMNH** | Zhejiang Museum of Natural History, Hangzhou, China |

# ACKNOWLEDGEMENTS

For access to specimens under their care, C.S. thanks Fangfang Teng (XHPM); R.V.P thanks Xiaolin Wang (IVPP), Shunxing Jiang (IVPP), Dieter Schreiber (SMNK), and Eberhard Frey (SMNK); and X.Z. thanks Xinsheng Jin (ZMNH), Qiannan Zhang (BMNH), Shaowen Zhang (CAGS), Jun Zhang (BXGM), Honggang Huo (BXGM), and Deyu Sun (JPM). R.V.P thanks Maria E. Leal, Lucy Souza and Kamila Bandeira for fruitful discussions.

## Funding

This work was supported by the Nature Science Foundation of Liaoning (No. 2019-MS-105), the FAPESP for a scholarship (#2019/10231-6), the Slovak Research and Development Agency (APVV-18-0251), the Scientific Research Foundation of Hainan Tropical Ocean University (No. RHDRC202008), the National Natural Science Foundation of China (grant #41688103, #41790452), and the China Geological Survey (DD20190397). The funders had no role in study design, data collection and analysis, decision to publish, or preparation of the manuscript.

## Grant Disclosures

The following grant information was disclosed by the authors:
The Nature Science Foundation of Liaoning: No. 2019-MS-105.
The FAPESP for a scholarship: #2019/10231-6.
The Slovak Research and Development Agency: APVV-18-0251.
The Scientific Research Foundation of Hainan Tropical Ocean University: No. RHDRC202008.
The National Natural Science Foundation of China: #41688103, #41790452.
The China Geological Survey: DD20190397.

## Competing Interests

The authors declare there are no competing interests.

## Author Contributions

- Caizhi Shen, Rodrigo V. Pêgas, Chunling Gao, Martin Kundrát, Lijun Zhang, Xuefang Wei and Xuanyu Zhou conceived and designed the experiments, performed the experiments, analyzed the data, prepared figures and/or tables, authored or reviewed drafts of the paper, and approved the final draft.

## Data Availability

The raw measurements are available in Table 1.

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
