# Peer review of "A new specimen of Sinopterus dongi (Pterosauria, Tapejaridae) from the Jiufotang Formation (Early Cretaceous, China)"

_PeerJ, doi:10.7717/peerj.12360_

## Round 0.1 · original submission · Minor Revisions

Please, together with your unmarked revised manuscript, provide a marked-up copy as well as a document explaining how you have addressed every issue raised by the reviewers.

·

Basic reporting

The article has a clear structure, with logical sub-sections, appropriate detailed, written in accessible/direct language and annotated figures complementing the text.
The raw data is shared well, the photographs are of high quality and focus on areas of interest described in the text. Specimen descriptions tend to lack detailed photographic pieces of evidence to back up the observations - but this paper does it really well, with well-annotated and outlined companion pieces which make reading more accessible.

I would suggest adding measurements of other elements (ie manual digits) to Table 1. Suggestions on additions to figures and literature used are included in the "Experimental design" & "Validity" sections.

The paper is self-contained, basing comparative description and conclusions only within clades of relation to the specimen.

Experimental design

The paper is tackling a hotly debated topic within pterosaur phylogeny, which is the Sinopterus taxonomic-over splitting, helping to solve this problem providing one of the more detailed osteology descriptions of Sinopterus. And it is refreshing and much needed to see a specimen description that is not another taxa assigned on poorly defined autapomorphies!

There are minor corrections I would like to propose to increase rigidity of the observations.

As the paper relies on comparative description, more comparable numerical values would be useful and help with cross-referencing, in example 146 line “These vertebrae are relatively elongate slightly more so than in Tapejara wellnhoferi or thalassodromines”, if possible use of comparative ratios might be useful and more definitive. If feasible, comparative illustrations complementing the comparisons might be useful (ie wingspan and anatomy diagram to aid visualisation of ontogenetic variation noted on lines 336-340) or illustrating humeral morphology differences as described in lines 393-403. As of now, the most comparative section is qualitative, the quantitative approach might be useful (although I am aware this might not always be the best approach when describing pterosaur elements). Comparing pterosaur remains is hard, as you frequently need to resort to features that are preserved at different angles, in poor quality, in 3D or flattened – making a robust assessment of features (especially humeral or scapula-coracoidal) hard. But the paper and its well-annotated illustrated outlines of elements accompanying the photographs mitigate this problem (especially aiding resolution of trunk elements in Figure 3). Some of this uncertainty could be mentioned, especially when comparing elements poorly preserved like the scapula.

The manual claw preservation, especially the keratin is really interesting considering how rarely it is found (and how strongly accentuated it is in this specimen). Describe in more detail (to line 288-289) on how pedal and ungual claws differ, while line 318 states " manual claws have the same shape", manual phalanges have somewhat visibly larger flexor processes. While full discourse into the subject lies outside the aims of this paper, higher resolution in the description of these elements might be beneficial!

Validity of the findings

The concluding observations in assessment and comparison of the specimen are valid and robust, with minor suggestions for improvements on conclusive remarks.

As pneumatic openings are an important aspect of tapejarid taxonomic assignment, evolution and description of this paper – how are they assessed to make sure openings are indeed pneumatic? Especially with a severe degree of flattening in regions assigned as bearing foramina. How certain are you with the assignment of pneumatic opening over a fracture or non-pneumatic depression within the bone?

As ontogeny is an important aspect of this papers discussion body and I agree there is sufficient information (line 326-328) to show the specimen is indeed immature. The cited text is slightly out of date and sources used to assign ontogenetic stage disputed by the community – and in the case of this specimen – not fully transferrable. The ontogenetic stages proposed by Kellner (2015) applied to basal, much smaller non-pterodactyloid of the Triassic. Dalla Vecchia (2018) wrote a paper disputing this ontogeny sequence scheme (https://doi.org/10.13130/2039-4942/10099), to quote “Kellner’s six ontogenetic stages are an oversimplification of a very complex process. Ontogenetic features of different taxa that probably had distinct growth patterns have been grouped together into a sequence that Kellner (2015) considered being valid for all pterosaurs. (…) It is evident that the order of fusion of composite skeletal elements during ontogeny differs in pterodactyloids and early pterosaurs and there is no universal pattern that can be extrapolated to pterosaurs in general.” There is also this fairly recent summary paper on this topic that might be of use (https://doi.org/10.1111/brv.12666). While Kellners OS is frequently used in literature, especially in the context of clade investigated in this paper, I would suggest refraining from using OS stages or at least providing a caveat on why this framework might be used with hesitancy.

Additional comments

I am eagerly awaiting a description of more than a dozen specimens currently in storage (mentioned on line 56), as these might bring much-needed clarity & look forward to more research on derived pterosaur phylogeny this paper will springboard!

·

Basic reporting

No comment

Experimental design

No comment

Validity of the findings

No comment

Additional comments

The present work is concerned with providing a detailed osteological description of a new specimen that, in my view and in agreement with the authors, is unequivocally a Sinopterus dongi. Overall, the article is solid and very well written.
I agree with the publication of this manuscript, since the description of these post-cranial elements has the potential to resolve biological, ontogenetic and taxonomic inconsistencies related to that genus. I do have a few comments that the authors should use as they see fit.

1 – Line 149 – I think that presenting a formula the length of a cervical series preserved in articulated and flattened might be inappropriate. What criteria were used for this measurement? from the most anterior to the most posterior point of the vertebrae?
Furthermore, according to Table 1, the cervical vertebra IV is not the longest in the series, as the length formula shows.

2 – Line 156 – Establishing a relation by height of the neural spine may not be appropriate due to the deformation of the flat material, since, according to Table 1, this seems to vary randomly throughout the cervical series due to preservation. Perhaps it is more appropriate to establish this relationship with the form of this structure.

3 – Line 165 – I have seen Thalassodrominae pterosaur vertebrae with 1, 2, or 3 lateral pneumatic foramina. The number of penumatic foramina often varies between individuals, vertebrae, and even sides of the vertebra. I think it would be more appropriate if the authors limited themselves to the presence or absence of foramina on the side and not if these are single or double foramina

4 – Line 192 – Here too I would not bother to include data on the height of neural spines of the dorsal vertebrae, which besides being crushed are apparently broken. The measures are also not present in Table 1

5 – Line 193 – The fossae appear to be present in a biomechanically appropriate region and closely resemble the pneumatic fossae found in the dorsal vertebrae of some Rhamphorhynchidae. However, due to the preservation of the material I find it inappropriate to state this in a description.

6 – Line 226 – Is the diameter measurement of the elements preserved in these conditions reliable?

7 – Line 342 – What do you mean "straight femoral shaft"? The femur is not described

8 – Line 413 – Here again I am concerned if the presence of two pneumatic foramina could be associated with an individual variation

9 – Figure 5 – In the figure legend, it says “A” and “B” where it should be “C” and “D”.

10 – Table 1 – Which unit of measure is being used in the values shown in Table 1? mm?

Review by Richard Buchmann

·

Basic reporting

Dear,

Below are my comments.

In this paper is presented in this the complete description of a new postcranial specimen of Sinopterus dongi (D3072) preserved on a single slab from the Cretaceous Jiufotang Formation. The skeleton is missing the skull, but shows confidential data, which permits the correct assignation to that tapejarine species.

The Tapejarinae represents edentulous pterosaurs with a considerable diversity during the Cretaceous. According the authors, for the Jiufotang Formation of China, a total of 7 nominal tapejarid species and 2 genera have been proposed. Despite the abundance of specimens (described and undescribed) and the relevant taxonomic problems involving of the tapejarine pterosaur Sinopterus dongi, detailed revisions of the matter were still lacking.

The manuscript is clearly written in professional, unambiguous language. It is very interesting, particularly because discuss about the problematic of the Sinopterus complex, with the addition of new information on the comparative osteology.

I hope that I can be helpful with this contribution.

Best regards,

Hebert Bruno Campos

Experimental design

Original primary research within Aims and Scope of the journal.

The investigation must was conducted rigorously and to a high technical standard. The research was conducted in conformity with the prevailing ethical standards in the field (Vertebrate Paleontology).

Validity of the findings

Conclusions are well stated, linked to original research question, supporting the results.

Additional comments

In addition, I suggest the authors to submit one single graphic with the main measurements of the limbs elements of the specimens of the Sinopterus complex, inclusing the new one, and another tapejarine species, to demonstrate the metric variation of the limb proportions.

---

## Round 0.2 · accepted · Accept

I confirm that your article has been accepted for publication.

·

Basic reporting

No comment

Experimental design

No comment

Validity of the findings

No comment

Additional comments

Dear editor,

The work is well written, direct and cohesive. The description of this material is extremely relevant to resolve taxonomic inconsistencies in the clade. All my doubts raised were clarified. I am satisfied with the modifications and responses given by the authors of the manuscript and suggest the publication of this work.